# Use of Almond Skins to Improve Nutritional and Functional Properties of Biscuits: An Example of Upcycling

**DOI:** 10.3390/foods9111705

**Published:** 2020-11-20

**Authors:** Antonella Pasqualone, Barbara Laddomada, Fatma Boukid, Davide De Angelis, Carmine Summo

**Affiliations:** 1Department of Soil, Plant and Food Science (DISSPA), University of Bari Aldo Moro, Via Amendola, 165/a, I-70126 Bari, Italy; davide.deangelis@uniba.it (D.D.A.); carmine.summo@uniba.it (C.S.); 2Institute of Sciences of Food Production (ISPA), CNR, via Monteroni, 73100 Lecce, Italy; barbara.laddomada@ispa.cnr.it; 3Institute of Agriculture and Food Research and Technology (IRTA), Food Safety Programme, Food Industry Area, Finca Camps i Armet s/n, 17121 Monells, Catalonia, Spain; fatma.boukid@irta.cat

**Keywords:** almond skins, by-product, upcycling, biscuits, health claims, fiber, nutritional composition, sensory properties, phenolic compounds

## Abstract

Upcycling food industry by-products has become a topic of interest within the framework of the circular economy, to minimize environmental impact and the waste of resources. This research aimed at verifying the effectiveness of using almond skins, a by-product of the confectionery industry, in the preparation of functional biscuits with improved nutritional properties. Almond skins were added at 10 g/100 g (AS10) and 20 g/100 g (AS20) to a wheat flour basis. The protein content was not influenced, whereas lipids and dietary fiber significantly increased (*p* < 0.05), the latter meeting the requirements for applying “source of fiber” and “high in fiber” claims to AS10 and AS20 biscuits, respectively. The addition of almond skins altered biscuit color, lowering *L** and *b** and increasing *a**, but improved friability. The biscuits showed sensory differences in color, odor and textural descriptors. The total sum of single phenolic compounds, determined by HPLC, was higher (*p* < 0.05) in AS10 (97.84 µg/g) and AS20 (132.18 µg/g) than in control (73.97 µg/g). The antioxidant activity showed the same trend as the phenolic. The *p*-hydroxy benzoic and protocatechuic acids showed the largest increase. The suggested strategy is a practical example of upcycling when preparing a health-oriented food product.

## 1. Introduction

Recently, the reuse of food industry by-products has become a particularly important research topic, in order to develop systems capable of minimizing environmental impact and the waste of resources. The confectionery industry, in the production of blanched almonds, generates large quantities of almond skins as a by-product, which are mostly destined to cattle feeding [1] and composting [2]. However, almond skins can be considered functional food ingredients because they contain several bioactive phenolic compounds, namely flavonoids, phenolic acids, and tannins, the latter both hydrolysable and condensed [3,4,5,6,7]. The phenolic content of fresh almond skins comprises between 11.1 and 17.7 mg/g, depending on the extraction protocol [7], whereas 0.25–0.85 mg/g d.m. (dry matter) were quantified in dried almond skins, with the lowest amount in sun-dried skins and the highest in skins oven-dried at a temperature of 45–60 °C [7].

The polyphenols of almond skins are bioavailable and possess in vitro and in vivo antioxidant activity, able to reduce plasmatic oxidative stress [8] and to protect LDL (low-density lipoprotein) from oxidation [4,9]. The bioactive compounds of almond skins display also antibacterial and antiviral effects [10,11]. Recently, an extract of almond skins has been proposed for use in intestinal inflammatory diseases [12]. Furthermore, almond skins are also a rich source of fiber and therefore have a prebiotic effect, favorably influencing the gut microbiome [13,14]. The recommended daily intake of fiber ranges from 18 g to 38 g for adults and it varies among different countries, but many people do not reach this threshold [15]. Almond skins could hence be used to functionalize foods and to improve their nutritional profile in terms of fiber content. The reuse of almond skins in food products would represent an example of upcycling [16], responding to the need to increase sustainability in the food industries within the framework of the principles of a circular economy [17].

Functional ingredients, such as almond skins, could be easily added to cereal-based products, but any modification of the physico-chemical and sensory characteristics of the end-products should be carefully evaluated so as to fulfill consumer expectations for healthy but pleasant foods. The potential use of almond skins in composite dough with wheat flour was evaluated in a previous study, highlighting significant alterations of alveograph and farinograph indices due to the presence of fibers, which interfere with the gluten network [7]. Therefore, almond skins could be used in those cereal-based products which better tolerate a weak gluten network, such as biscuits.

Biscuits are popular baked goods, eaten daily and characterized by a long shelf-life. These features make biscuits a good recipient for the addition of functional ingredients. To date, however, almond skins are still an underexploited resource and no study has considered their introduction in biscuit formulation, despite many researchers having reformulated biscuits by incorporating an array of new ingredients, mostly of vegetable origin, such as apple peel powder [18], acorn flour [19], grape marc extract [20,21], purple wheat flour [22], inulin [23], soy protein isolate [24], blue berry by-product [25], and green tea extract [26].

Within this framework, the aim of this research has been to verify the effectiveness of almond skin addition in the preparation of functional biscuits with improved nutritional properties.

## 2. Materials and Methods

### 2.1. Raw Materials

The ingredients used for preparing the experimental biscuits were: refined wheat flour (0.52 g/100 g ashes) (Molini Spigadoro, Bastia Umbra, Italy), sucrose (Eridania, Bologna, Italy), extra virgin olive oil (Olearia De Santis, Bitonto, Italy), baking powder (sodium bicarbonate and potassium bitartrate, ‘Belbake’, Lidl Stiftung & Co. KG, Neckarsulm, Germany), all purchased at local retailers, and almond skins. The latter were collected from an almond processing industry (Calafiore S.r.l., Floridia, Italy), then dried at 60 °C for 30 min by a rotary air drier (mod. Scirocco, Società Italiana Essiccatoi, Milano, Italy), milled (Cutting Mill SM 100, Retsch, Haan, Germany) and sieved on a sieve with 0.6 mm holes. Moisture, a_w_, phenolic compounds, antioxidant activity, color, and odor notes of almond skins are reported in a previous paper [7].

### 2.2. Preparation of Biscuits

The formulation of biscuits is reported in Table 1. Two levels of addition of almond skins were considered: 10 g/100 g (AS10) and 20 g/100 g (AS20) on a wheat flour basis, which were compared with control biscuits prepared without adding almond skins. The amount of water was defined in preliminary trials in order to achieve the same dough workability in the three types of biscuits. The process consisted in: kneading for 3 min sucrose, extra virgin olive oil and baking powder by an electric mixer with flat beater (Kitchen Aid, Antwerp, Belgium), then adding flour (pure wheat flour or blended with almond skin powder as in Table 1) and kneading for 3 min, finally adding water and kneading for about 10 min to form a homogeneous dough. The dough was then rolled out with a rolling pin to a thickness of 4 mm and cut into 6 cm diameter disks with the aid of a circular biscuit cutter with scalloped edges. The disks of dough were placed on a baking tray, mixing them in a randomized block pattern to minimize any effect of tray location during baking, then were baked in an electric oven (mod. Ignis ACF961IX, Whirlpool Italia S.r.l., Pero, Italy) at 175 °C for 15 min. Two independent production trials were carried out. Biscuits were finely crushed for analysis, except for the textural, colorimetric and sensory analyses.

### 2.3. Determination of Nutritional Composition

Protein (N × 5.7) and moisture content were determined according to the American Association of Cereal Chemists (AACC) Methods 46–11.02 and 08–01, respectively [27]. The lipid fraction was extracted according to ICC Standard Method no. 136 [28]. Total dietary fiber was determined by the enzymatic-gravimetric procedure according to the AOAC Official Method 991.43 [29]. Carbohydrates were calculated by difference: 100 – (moisture + proteins + lipids + fiber + ash). Energy value (kJ), calculated by using the Atwater general conversion factors, also considered the contribution of 8 kJ/g from total dietary fiber, according to Annex XIV of Regulation (EC) No 1169/2011 [30]. All analyses were carried out in triplicate.

### 2.4. Determination of Physical Properties

The *a** (red/green balance), *b** (yellow/blue balance), and *L** (lightness) coordinates of the CIELAB color space were determined by a colorimeter (CM-600d Chromameter, Konica Minolta, Tokyo, Japan) under illuminant D65. Five replicated analyses were carried out. Total color difference (ΔE) was calculated as follows [31]:

∆E = [(∆*L**)^2^ + (∆*a**)^2^ + (∆*b**)^2^]^1/2^

The following scale was considered: ΔE = 0–0.5, very low difference; 0.5–1.5; slight difference; 1.5–3.0, noticeable difference; 3.0–6.0, appreciable difference; 6.0–12.0, large difference; and >12.0, very obvious difference [32].

Water activity (a_w_) was analyzed in triplicate by a water activity meter (mod. Aqualab 4TE, Meter group, Pullman, WA, USA).

Textural properties, in terms of breaking strength (N mm^−2^), were determined by a three-point bending test (“snap test”) using a ZI.0 TN texture analyzer (ZwickRoell GmbH & Co. KG, Ulm, Germany), equipped with 1 kN load-cell. The biscuits were placed on the analyzer supports with their top surface down. The distance between the support bars was 4 cm. The downward movement of the probe, set at a speed of 5 mm min^−1^, was continued until the biscuit was broken. Eight replicated analyses were carried out.

### 2.5. Baking Induced Variations of Dimensional Parameters and Weight

The weight (W) of biscuits before and after baking was assessed by a balance (Gibertini, Novate Milanese, Italy). The diameter (D) and thickness (T) of biscuits before and after baking were determined by a caliper. The spread factor was calculated as the ratio between D and T of baked biscuits, according to the AACC Method 10-50.05 [27]. The percentage variations in W, D, and T were calculated as follows:

% variation of W (or D, T) = (W (or D, T) after baking—W (or D, T) before baking)/W (or D, T) before baking × 100. Six replicated analyses were carried out.

### 2.6. HPLC analysis of Phenolic Compounds

The phenolic compounds were extracted from 1 g biscuits according to the procedure reported in Laddomada et al. [33], which involved defatting, alkaline hydrolysis, acidification and double ethyl acetate extraction. The extracts were lyophilized and dissolved in 400 µL of a solution of methanol diluted with 200 mL/L distilled water, then 50 μL were filtered on 0.45 μm polytetrafluoroethylene (PTFE) filters (Teknokroma, Barcelona, Spain) and analyzed by HPLC-DAD (Agilent 1100 Series, Agilent Technologies, Santa Clara, CA, USA) with a reversed phase C18(2) Luna column (Phenomenex, Torrance, CA, USA) (5 μm, 250 × 4.6 mm), as in Pasqualone et al. [7]. Identification of peaks was made by comparison of their UV-Vis spectra, and their retention times to those of authentic phenolic standards. Phenolic acids were quantified via a ratio of 3,5-dichloro-4-hydroxybenzoic acid, used as internal standard, and calibration curves of phenolic acid standards. Other phenolics (flavan-3-ols, flavonol and flavonone glycosides and aglycones) were quantified using calibration curves according to the external standard method [6]. The linear range, correlation coefficient, limit of detection (LOD) and limit of quantification (LOQ) for the phenolic compounds quantified are reported in Appendix A.

### 2.7. Determination of Antioxidant Activity

An amount of 1 g sample, mixed with 10 mL of methanol and shaken at 250 rpm for 2 h in the dark, was centrifuged for 5 min at 5000 × g. The supernatant was submitted to the assessment of the antioxidant activity by the 2,2-diphenyl-1-picrylhydrazyl (DPPH) radical scavenging capacity assay, as in Pasqualone et al. [22]. A calibration curve was prepared with 0.1–100 μM solutions of 6-hydroxy-2,5,7,8-tetramethylchroman-2-carboxylic acid (Trolox) (Sigma–Aldrich Chemical Co., St. Louis, MO, USA) (y = −0.008x + 0.6087; *R^2^* = 0.9971).

### 2.8. Determination of Sensory Properties

Quantitative Descriptive Analysis (QDA) of biscuits was performed by a trained sensory panel of eight people, following the ethical guidelines of the laboratory of Food Science and Technology of the Department of Soil, Plant and Food Science (DISSPA), Dept. of Bari University (Italy). Panelists, regular consumers of biscuits and almonds and free of food intolerances or allergies, were informed about the study aims, and signed an individual written informed consent. Pre-test sessions were carried out, as in Pasqualone et al. [34]. Eight sensory descriptors, defined in Table 2, were rated on a 0-9 score range (0 = minimum; 9 = maximum intensity). The analyses were carried out in triplicate.

### 2.9. Statistical Analysis

One-way analysis of variance (ANOVA) followed by Tukey’s HSD test, was made using the XLSTAT software (Addinsoft SARL, New York, NY, USA).

## 3. Results and Discussion

### 3.1. Nutritional and Technological Characteristics

Almond skin powder is particularly rich in fiber (52.6 g/100 g), as shown by the analysis of its nutritional characteristics (Table 3).

This by-product of almond processing also showed a relevant presence of lipids (21.3 g/100 g). The lipid fraction of almond skins, however, is particularly healthy, being composed mainly of mono and polyunsaturated fatty acids (mostly oleic and linoleic acids) [6] associated with high amounts of vitamin E [6]. The composition of the lipid fraction of skins parallels the lipid composition of the whole seed [35]. The protein content of almond skins accounted for about 11 g/100 g, and low amounts of carbohydrates were observed. The overall composition of almond skin powder agreed with the current literature [6,36]. The composition of wheat flour was quite different than that of almond skins, being rich in carbohydrates and poor in fiber, with negligible levels of lipids.

The analysis of the nutritional features of biscuits (Table 4) shows that the protein content was not significantly influenced by the addition of almond skins, the latter having a protein content similar to wheat flour. However, AS20 biscuits had a significantly (*p* < 0.05) higher lipid content than control, due to the relevant contribution of almond skins. The lipid content of all biscuits was in the range of those commonly marketed [37].

As for the content of dietary fiber, it progressively increased with the increase of almond skin addition. EC Regulation n. 1924/2006 [38], relating to nutrition and health claims made on food products, defines that a food is a “source of fiber” only if contains at least 3 g/100 g fiber, or at least 1.5 g/100 kcal fiber, while “high in fiber” applies only if a food contains at least 6 g/ 100 g fiber, or at least 3 g/100 kcal fiber. The level of fiber ascertained in AS10 and AS20 biscuits met the requirements for applying the “source of fiber” and the “high in fiber” claims, respectively.

Moisture content increased, but not significantly, after the addition of almond skins due to their contribution of fiber. The higher the protein and fiber content, the higher the water absorption by the dough and moisture retention are found of the final product [39].

As a consequence of the increase in fats and fiber, the level of carbohydrates significantly decreased in almond skin-added biscuits compared to control. The energy value did not vary significantly by adding almond skins, because the increase of lipids was compensated for by an increase of fiber and a decrease in carbohydrates.

As for the main physical characteristics (Table 5), the a_w_ of AS10 and AS20 was slightly higher than control, but without a significant difference. The a_w_ values observed in all biscuits agreed with moisture content and showed that they were conveniently dry and stable from the microbiological point of view (a_w_ < 0.6).

The addition of almond skins, which were brown colored, resulted in an expected substantial alteration of biscuit color (Figure 1), with a significant decrease of *L** and *b**, and an increase of *a** in AS10 and AS20 compared to the control (Table 5). The total color difference (∆E) of AS10 and AS20 biscuits compared to the control was greater for AS20 than for AS10, but in both cases with very high values, confirming that the control had a distinct color [40]. ∆E values >12.0, in fact, indicate a very obvious color difference [32].

The textural analysis showed that the addition of almond skins caused a decrease in the strength necessary to break the biscuits, i.e., an increase of friability, which is a particularly important characteristic. Friability is a salient textural characteristic for biscuits [41,42]. Tough, non-crumbly biscuits have low acceptance values in consumer tests [43]. This variation of breaking strength was significant when comparing control with AS20 and was due to the high presence of fiber in the almond skins. Fibers are highly hygroscopic and interfere with the formation of a strong and complete gluten network [44]. Preliminary work, in fact, showed that the rheological properties of the dough [7] significantly worsened after the addition of almond skin powder. However, among baked goods, biscuits are the most suitable for being reformulated with the addition of fibrous raw materials, since for their production a weak gluten network is not only sufficient but even necessary. In addition, although the difference in lipid content with control was significant only for AS20, the lipid fraction deriving from almond skins could have positively influenced the friability [45]. Therefore, the addition of almond skins did not harden biscuits at all; on the contrary, it gave a crumblier texture.

As for the dimensional variations induced by baking (Table 6), due to the thermal expansion of gases (carbon dioxide developed by the baking powder, dough moisture, and air entrapped during kneading), all biscuits increased more in thickness than in diameter. This result, commonly observed in biscuit baking [19], is due to the retaining effect of gluten, which tends to limit enlargement, whereas the upward thrust of the oven heat (oven rise) is less opposed [45]. AS10 and AS20 showed a greater diameter increase than control, which was significantly different for AS20, but had a lower increase in thickness. The easier enlargement observed in almond-skin added biscuits was due to the coupled effect of the dilution of gluten by a non-gluten raw material and the interference with gluten formation by the fiber and lipids of the same material. These findings agreed with studies where other fibrous and gluten-free ingredients were added to biscuit dough [18,19]. In addition, better expanded biscuits are usually less compact and more friable than those which expand less, in agreement with the observed textural data.

The spread factor increased progressively as the amount of almond skins increased, with a significant difference between control and AS20. A higher spread factor indicates a better quality and is linked to an increase in consumer acceptability [46]. The observed values were higher than those reported for biscuits enriched with pure fiber of various cereals [47].

Weight loss, primarily due to the moisture loss from dough during baking, decreased by increasing the amount of almond skins as a consequence of the greater hygroscopicity of fibers, which limited water migration. The values ascertained were in the range of other researches [48,49,50].

The sensory profiles of the biscuits showed significant differences in odor, color and textural descriptors (Table 7). As for taste, the bitter note was negligible in the biscuits investigated, while sweetness was moderately intense, both without significant difference among formulations.

A slight odor note of caramel, derived from sugar caramelization and Maillard reaction, was perceived by the panelists in all biscuit types, without statistically significant differences between them. Instead, differences between the samples were found in the intensity of leafy odor. This odor note, absent in the control, was perceived with low intensity in biscuits formulated with almond skins, with the highest perception in AS20 and with an intermediate value in AS10. In previous research [7] this characteristic smell note was observed in the dried almond skins used in biscuit-making, albeit much more pronounced than in the finished product.

The color of biscuits became progressively and significantly darker as the level of addition of almond skin powder increased, as already indicated by colorimeter determinations.

As for friability, evaluated as the way biscuit fractured when broken by finger, the sensorial results were similar to those obtained instrumentally by the texture analyzer (snap test). AS20 was significantly more friable than control.

Dryness and graininess, on the other hand, were evaluated during chewing. Dryness did not show significant differences, whereas graininess was scored higher in almond-skin added biscuits, due to their granular and fibrous crumbles.

### 3.2. Functional Characteristics

Almond skins are rich in phenolic compounds [7], therefore the content of these bio-actives was evaluated in biscuits, as well as antioxidant activity (Table 8). The total sum of phenolic compounds, determined by HPLC, was significantly higher in AS10 and AS20 than in control.

In more detail, the variation of the single phenolics did not show the same trend for all the compounds, which showed different behavior according to the phenolic composition of the raw materials. In particular, among the phenolic acids, the *p*-hydroxy benzoic and protocatechuic acids showed a relevant increase after the addition of almond skins. The flavan-3-ols catechin and epicatechin also followed the same trend, being not detectable in control and showing a concentration-effect increment between the AS10 and AS20. In fact, these phenolic compounds are the most abundant in almond skins [7]. A smaller increase, but always statistically significant, was observed for syringic acid, vanillic and *p*-coumaric acids.

Instead, the most abundant phenolic acid, namely the ferulic acid, followed by the sinapic acid, decreased when comparing control biscuits with the almond-skin added, because these phenolic acids are typically present in wheat [33,51], but not in almond.

The flavonol glycosides and their aglycones, as well as the flavanone glycosides and their aglycones, despite their presence in almond skins [7], were not detected in biscuits. Probably, since their starting quantity was not remarkably high, they became undetectable in the biscuits, due to the dilution effect of wheat flour. In addition, oxidation and other degradation phenomena could not be excluded during processing (kneading and baking) since a decrease in phenolic compounds had already been observed when raw almond skins were thermally dried [7]. In any case, the total phenolic compounds of AS20 were approximately double that of the control, indicating that the addition of almond skins in the formulation can concretely contribute to enhance the nutritional value and the potential health benefits of the end products.

The antioxidant activity followed the same trend as the phenolic and showed higher values in the almond skin supplemented biscuits, compared to the control, also highlighting a concentration effect. Indeed, in the AS20, the antioxidant activity was about five times higher than the control. The observed values of antioxidant activity were consistent with those of the almond skins added [7].

## 4. Conclusions

The increasing sensibility of modern consumers towards the potential benefits of food on human health has led to a strong demand for functional products.

To date, almond skins, in spite of having high fiber content and antioxidant substances, are a by-product of almond processing usually addressed to animal feed and/or composting. This study, instead, demonstrates that almond skins can be effectively used for the production of functional biscuits, addressing the needs of both producers, who require the reduction of waste production, and consumers, who increasingly demand healthier food. For this latter purpose, the nutritional claims “source of fiber” and “high in fiber”, defined in EC Regulation n. 1924/2006, were applicable to the AS10 and AS20 biscuits, respectively.

Therefore, using almond skins in biscuit-making is a feasible way to convert a low-value by-product into a valuable resource, providing to the almond processing industry an efficient and environment-friendly solution for waste disposal. This is a practical example of upcycling while preparing a health-oriented food product.

## Figures and Tables

**Figure 1 foods-09-01705-f001:**
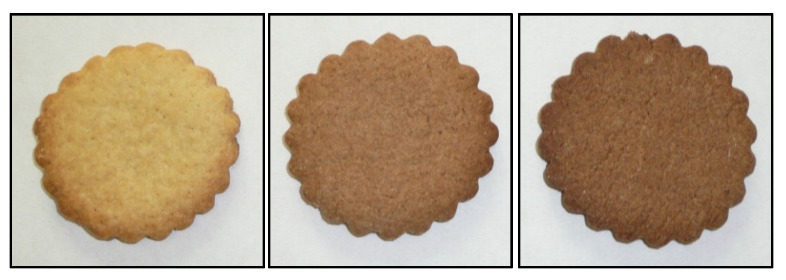
Biscuits enriched by increasing levels of almond skins. From left to right: Control = biscuits prepared without adding almond skins; biscuits prepared by adding 10 g of almond skin powder per 100 g of wheat flour (AS10); biscuits prepared by adding 20 g of almond skin powder per 100 g of wheat flour (AS20).

**Table 1 foods-09-01705-t001:** Formulation of the experimental biscuits (per 100 g of flour). Control = Biscuits without Almond Skins; AS10 and AS20 = Biscuits prepared by adding 10 g and 20 g Almond Skin Powder per 100 g of Wheat Flour, respectively.

	Control	AS10	AS20
Wheat flour (g)	100	90	80
Almond skin powder (g)	-	10	20
Sucrose (g)	28	28	28
Extra virgin olive oil (g)	18	18	18
Water (g)	26	28	30
Baking powder (g)	1	1	1

**Table 2 foods-09-01705-t002:** Descriptive terms used for the sensory profiling of biscuits.

Descriptor	Definition	Scale Anchors	
		Min (0)	Max (9)
*Odor*			
Caramel odor	Typical odor associated with caramel	Absent	Very intense
Leafy odor	Smell reminiscent of green leaves	Absent	Very intense
*Visual-tactile characteristics*			
Color	Color of biscuit surface	Beige	Dark brown
Friability	The way the biscuit fractures, when broken by fingers	Very tough, it breaks with difficulty	Very friable and crumbly, it breaks easily
*Taste*			
Sweetness	Basic taste produced by sucrose	Absent	Very intense
Bitterness	Basic taste produced by caffeine	Absent	Very intense
*Texture attributes perceived during chewing*			
Dryness	Dryness perceived at the surface of biscuit	Moist	Very dry
Graininess	Graininess perceived at the end of chewing	Not grainy, giving finely sized crumbs	Very grainy, giving differently sized crumbs, medium and large

**Table 3 foods-09-01705-t003:** Nutritional composition of dried almond skin powder and wheat flour used in the preparation of experimental biscuits. Values per 100 g, expressed on fresh weight basis.

	Almond Skin Powder	Refined Wheat Flour
Moisture (g)	10.1 ± 0.2	14.2 ± 0.4
Carbohydrates (g)	5.4 ± 0.5	73.3 ± 0.9
Fats (g)	21.3 ± 0.6	0.9 ± 0.1
Proteins (g)	10.6 ± 0.2	9.8 ± 0.2
Fiber (g)	52.6 ± 0.5	1.8 ± 0.2

**Table 4 foods-09-01705-t004:** Nutritional features (values per 100 g, expressed on fresh weight basis) of biscuits enriched by increasing levels of almond skins. Control = biscuits without almond skins; AS10 and AS20 = biscuits prepared by adding 10 g and 20 g of almond skin powder per 100 g of wheat flour, respectively.

	Control	AS10	AS20
Moisture (g)	5.2 ± 0.3 a	5.5 ± 0.3 a	5.6 ± 0.4 a
Carbohydrates (g)	77.8 ± 1.1 a	74.3 ± 1.2 b	70.2 ± 0.7 c
Fats (g)	10.3 ± 0.9 b	11.5 ± 0.4 a,b	12.4 ± 0.4 a
Proteins (g)	5.6 ± 0.1 a	5.6 ± 0.2 a	5.6 ± 0.3 a
Fiber (g)	1.1 ± 0.2 c	3.1 ± 0.1 b	6.2 ± 0.2 a
Energy value (kJ)	1794 ± 9 a	1797 ± 8 a	1789 ± 10 a

Different letters in row indicate significant differences (*p* < 0.05).

**Table 5 foods-09-01705-t005:** Physical characteristics of biscuits enriched by increasing levels of almond skins. Control = biscuits without almond skins; AS10 and AS20 = biscuits prepared by adding 10 g and 20 g of almond skin powder per 100 g of wheat flour, respectively.

	Control	AS10	AS20
a_w_	0.24 ± 0.02 a	0.27 ± 0.01 a	0.28 ± 0.02 a
*Colorimetric data*			
*a^*^*	9.85 ± 1.09 b	12.35 ± 0.43 a	12.61 ± 0.36 a
*b^*^*	35.12 ± 1.25 a	24.94 ± 0.66 b	22.66 ± 0.64 c
*L^*^*	68.7 ± 2.41 c	49.35 ± 0.85 b	46.15 ± 1.01 a
∆E_vs Control_	-	21.08 ± 0.51	25.76 ± 0.69
*Texture*			
Fracture strength (N/mm²)	8.82 ± 0.66a	7.63 ± 0.49ab	6.87 ± 0.31b

Different letters in row indicate significant differences (*p* < 0.05).

**Table 6 foods-09-01705-t006:** Baking induced variations of dimensional parameters of biscuits enriched by increasing levels of almond skins. Control = biscuits without almond skins; AS10 and AS20 = biscuits prepared by adding 10 g and 20 g of almond skin powder per 100 g of wheat flour, respectively.

	Control	AS10	AS20
Diameter variation (%)	3.45 ± 0.91 b	4.51 ± 0.72 a,b	6.21 ± 0.89 a
Thickness variation (%)	40.23 ± 3.41 a	38.12 ± 2.98 a	33.65 ± 1.04 b
Spread factor	10.83 ± 0.81 b	11.26 ± 0.38 a,b	12.45 ± 0.22 a
Weight loss (%)	15.04 ± 0.37 a	14.78 ± 0.43 a,b	14.09 ± 0.31 b

Different letters in row indicate significant differences (*p* < 0.05).

**Table 7 foods-09-01705-t007:** Sensory properties of biscuits enriched by increasing levels of almond skins. Control = biscuits without almond skins; AS10 and AS20 = biscuits prepared by adding 10 g and 20 g of almond skin powder per 100 g of wheat flour, respectively.

	Control	AS10	AS20
Caramel odor	2.3 ± 0.2 a	1.9 ± 0.1 a	2.1 ± 0.2 a
Leafy odor	0.0 ± 0.0 c	0.9 ± 0.2 b	1.6 ± 0.2 a
Color	4.3 ± 0.5 c	7.8 ± 0.4 b	8.9 ± 0.4 a
Friability	3.5 ± 0.2 b	3.7 ± 0.2 a,b	4.2 ± 0.3 a
Sweetness	4.4 ± 0.3 a	4.7 ± 0.2 a	4.6 ± 0.2 a
Bitterness	0.2 ± 0.1 a	0.1 ± 0.1 a	0.2 ± 0.1 a
Dryness	4.7 ± 0.3 a	4.8 ± 0.3 a	5.0 ± 0.4 a
Graininess	1.7 ± 0.1 b	2.5 ± 0.2 a	2.9 ± 0.3 a

Different letters in row indicate significant differences (*p* < 0.05).

**Table 8 foods-09-01705-t008:** Phenolic compounds and antioxidant activity of biscuits enriched by increasing levels of almond skins. Control = biscuits without almond skins; AS10 and AS20 = biscuits prepared by adding 10 g and 20 g almond skin powder per 100 g of wheat flour, respectively.

	Control	AS10	AS20
AA (DPPH) (µmol TE/g)	1.89 ± 0.16 c	6.11 ± 0.61 b	9.76 ± 0.74 a
*Single phenolic compounds* (µg/g)			
Vanillic acid	1.43 ± 0.02 c	2.77 ± 0.18 b	4.53 ± 0.10 a
Syringic acid	2.69 ± 0.02 c	6.04 ± 0.45 b	9.71 ± 0.30 a
*p*-Coumaric acid	0.36 ± 0.01 c	0.79 ± 0.12 b	1.14 ± 0.08 a
Ferulic acid	63.72 ± 0.52 a	55.21 ± 1.47 b	55.96 ± 1.50 b
Sinapic acid	5.33 ± 0.04 a	4.30 ± 0.15 b	4.25 ± 0.20 b
*p*-Hydroxybenzoic acid	0.44 ± 0.02 c	5.49 ± 0.19 b	12.96 ± 0.44 a
Protocatechuic acid	0.00 ± 0.00 c	3.55 ± 0.06 b	12.67 ± 0.31 a
(+)-Catechin	0.00 ± 0.00 c	11.17 ± 0.06 b	19.52 ± 1.06 a
(-)-Epicatechin	0.00 ± 0.00 c	8.54 ± 0.02 b	11.45 ± 0.21 a
Total sum	73.97 ± 0.54 c	97.84 ± 2.55 b	132.18 ± 1.63 a

AA = antioxidant activity; TE = Trolox equivalents; DPPH = 2,2-diphenyl-1-picrylhydrazyl radical. Different letters in row indicate significant differences (*p* < 0.05).

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
