# Peer review of "Use of Almond Skins to Improve Nutritional and Functional Properties of Biscuits: An Example of Upcycling"

_foods, 2020, doi:10.3390/foods9111705_

Round 1
Reviewer 1 Report
The paper addresses an important topic about the valorization of byproducts from the agri-food industry (almonds industrialization) to produce added-value products in this case healthy biscuits, contributing to the sustainability of the food chain. In this aspect, the work is very interesting and novel because this byproduct has been used mainly for feed animal and only a few studies have been found about its application for human food.
It is well written, with clear objectives, tables and figures fit the standard and the discussion is adequate.
However, I have some doubts about the following points:
Color evaluation: you have to clarify in which color space have you work?, because I don’t understand the way in which these data have been shown. In the case of the CIELAB color space, the a* and b* are coordinates: a* coordinate (red/green) and b* coordinate (yellow/blue) but you have named a* as red index and b* as yellow index and it is not correct. You have to include a reference for the brown index. If you want to compare how is the color differences between your samples and the control, the most adequate parameter would be the color differences (DE*)
Antioxidant properties: most authors agree than antioxidant activity can be developed by several mechanisms and that only one antioxidant method is not enough to measure this property and that is better than several antioxidant methods based on different antioxidant mechanisms can be assessed for a better evaluation. Why have you used only the DPPH method?
L225 “The textural analysis showed that the addition of almond skins caused a decrease of the strength necessary to break the biscuits, i.e. an increase of friability, which is a very appreciated characteristic”. This last statement must be referenced. Why friability is and appreciated characteristics instead of crunchy?
Author Response
Reviewer 1
The paper addresses an important topic about the valorization of byproducts from the agri-food industry (almonds industrialization) to produce added-value products in this case healthy biscuits, contributing to the sustainability of the food chain. In this aspect, the work is very interesting and novel because this byproduct has been used mainly for feed animal and only a few studies have been found about its application for human food.
It is well written, with clear objectives, tables and figures fit the standard and the discussion is adequate.
Response: We acknowledge the Reviewer for his/her positive evaluation of our research, and we thank him/her for careful reading and for helpful suggestions.
However, I have some doubts about the following points:
Color evaluation: you have to clarify in which color space have you work?, because I don’t understand the way in which these data have been shown. In the case of the CIELAB color space, the a* and b* are coordinates: a* coordinate (red/green) and b* coordinate (yellow/blue) but you have named a* as red index and b* as yellow index and it is not correct. You have to include a reference for the brown index. If you want to compare how is the color differences between your samples and the control, the most adequate parameter would be the color differences (DE*)
Response: Sorry for the lack of clarity. We worked in the CIELAB color space. This specification has been added to the text (see line 111).
Regarding the way we named the color indices (yellow index for b*, red index for a* and brown index for 100 – L*), there are several references where these color indices are named in this way (see explanation below). However, considering that this practical convention can be misleading for the majority of readers, we changed the manuscript according to the Reviewer’s suggestion and named the indices to remain adherent to the original CIELAB color space convention (i.e. simply a*, b* and L*) without any interpretations (see lines 110-113).
Furthermore, we calculated the DE vs. control (see Material and methods, lines 114-117) and inserted it in Table 5, thanks for suggesting. We commented it in the Results and discussion section (lines 228-231).
Please consider this explanation:
The way of naming a*, b* and L* as red, index and brown index (the latter for 100 – L*) is adopted for industrial quality checks in the mills and in bakery industries. In fact, the b* index (which, as you highlighted, of course is more correctly the yellow/blue coordinate, with positive values for yellow and negative for blue) for wheat-based products has usually a positive value, falling in the yellow area of the color space. Similarly, the a* coordinate, which is along the red/green variation, in cereal-based products has only positive values, falling in the red area. Finally, being the brownness a negative feature of cereal based products (negative from the commercial point of view, being related to decreased consumer appreciation) it is more usual to collect data of 100-L* (defined as brown index) than L* data (lightness).
References:
For “b*” as “yellow index” see: 1) Blanco, A., Colasuonno, P., Gadaleta, A., Mangini, G., Schiavulli, A., Simeone, R., ... & Cattivelli, L. (2011). Quantitative trait loci for yellow pigment concentration and individual carotenoid compounds in durum wheat. Journal of Cereal Science, 54(2), 255-264; 2) Borrelli, G. M., De Leonardis, A. M., Fares, C., Platani, C., & Di Fonzo, N. (2003). Effects of modified processing conditions on oxidative properties of semolina dough and pasta. Cereal Chemistry, 80(2), 225-231; 3) Parada, R., Royo, C., Gadaleta, A., Colasuonno, P., Marcotuli, I., Matus, I., ... & Schwember, A. R. (2020). Phytoene synthase 1 (Psy-1) and lipoxygenase 1 (Lpx-1) Genes Influence on Semolina Yellowness in Wheat Mediterranean Germplasm. International Journal of Molecular Sciences, 21(13), 4669.
For “a*” as “red index” see: 1) Marconi, E., Carcea, M., Schiavone, M., & Cubadda, R. (2002). Spelt (Triticum spelta L.) pasta quality: Combined effect of flour properties and drying conditions. Cereal Chemistry, 79(5), 634-639; 2) Cavazza, A., Corradini, C., Rinaldi, M., Salvadeo, P., Borromei, C., & Massini, R. (2013). Evaluation of pasta thermal treatment by determination of carbohydrates, furosine, and color indices. Food and Bioprocess Technology, 6(10), 2721-2731; 3) Abecassis, J., Faure, J., & Feillet, P. (1989). Improvement of cooking quality of maize pasta products by heat treatment. Journal of the Science of Food and Agriculture, 47(4), 475-485.
For “100-L*” as “brown index” see: Feillet, P., Autran, J. C., & Icard-Verniere, C. (2000). Mini review pasta brownness: an assessment. Journal of Cereal Science, 32(3), 215-233.
---
Antioxidant properties: most authors agree than antioxidant activity can be developed by several mechanisms and that only one antioxidant method is not enough to measure this property and that is better than several antioxidant methods based on different antioxidant mechanisms can be assessed for a better evaluation. Why have you used only the DPPH method?
Response: We adopted the DPPH method to compare the results of almond-skin added biscuits with published DPPH data of almond skins.
L225 “The textural analysis showed that the addition of almond skins caused a decrease of the strength necessary to break the biscuits, i.e. an increase of friability, which is a very appreciated characteristic”. This last statement must be referenced. Why friability is an appreciated characteristics instead of crunchy?
Response: The sentence has been reworded and referenced (lines 239-240).
At this regards, please consider the following explanation:
Crunchiness as well is very appreciated. Crispy and crunchy foods are appealing and enjoyable (Tunick, M. H., Onwulata, C. I., Thomas, A. E., Phillips, J. G., Mukhopadhyay, S., Sheen, S., ... & Cooke, P. H. 2013. Critical evaluation of crispy and crunchy textures: a review. International Journal of Food Properties, 16(5), 949-963). However, friability is a salient textural characteristic for biscuits (Piazza, L., Bringiotti, R., & Masi, P. 1994. The Influence of Moisture Content on the Failure Mode of Biscuits. In Developments in Food Engineering pp. 134-136. Springer, Boston, MA; Piazza, L., & Schiraldi, A. 1997. Correlation between fracture of semi‐sweet hard biscuits and dough viscoelastic properties. Journal of texture studies, 28(5), 523-541). Tough, non-crumbly biscuits have low acceptance values in consumer tests (Rojo-Poveda, O., Barbosa-Pereira, L., Orden, D., Stévigny, C., Zeppa, G., & Bertolino, M. 2020. Physical Properties and Consumer Evaluation of Cocoa Bean Shell-Functionalized Biscuits Adapted for Diabetic Consumers by the Replacement of Sucrose with Tagatose. Foods, 9(6), 814, doi:10.3390/foods9060814). Our aim was to verify that biscuits did not become too hard, dense and difficult to break.
Reviewer 2 Report
The content of the manuscript is relevant for the Research Area and the results are clear and well explained and discussed. Also, the English Grammar and style is fine. In my opinion, the only aspect that results less relevant is the novelty, as there are many articles on the use of bioactive aditives.
Author Response
Reviewer 2
The content of the manuscript is relevant for the Research Area and the results are clear and well explained and discussed. Also, the English Grammar and style is fine.
Response: We acknowledge the Reviewer for his/her positive evaluation of the manuscript.
In my opinion, the only aspect that results less relevant is the novelty, as there are many articles on the use of bioactive aditives.
Response: Yes, many papers consider the addition of bioactive compounds, and many of them consider the direct addition of bioactive food by-products and waste because there is the urgent need to turn our food industries to a more sustainable production. However, to the best of our knowledge no manuscript considered the re-use of almond skins in food products, and it has to be taken into account that the confectionery industry, in the production of blanched almonds, generates large quantities of almond skins as a by-product currently not properly reused.
Reviewer 3 Report
The article is well structured. The methodology used is adequate. The results are presented correctly and validated statistically.
Author Response
Reviewer 3
The article is well structured. The methodology used is adequate. The results are presented correctly and validated statistically.
Response: We acknowledge the Reviewer for his/her positive evaluation of our research.
Reviewer 4 Report
The article entitled: “Use of almond skins to improve nutritional and functional properties of biscuits: an example of upcycling” described the use of almond skins, an almond by-product, in the formulation of functional biscuits. The authors accessed how this addiction interfered in the proximate composition, color, friability, spread factor, phenolic profile, antioxidant activity and sensory properties. The work is well written, and it brings some interesting results and applications which are of interest to the Foods readers. I have some minor issues that need to be addressed:
Introduction: Line 51 : In the phrase : “The reuse of almond skins in food products would an example of upcycling [16],” something is missing, please revise it.
Lane 102: The same applies for Line Energy value (kJ) was calculated by Atwater general, please revise.
HPLC analysis of Phenolic Compounds: The authors are invited to add the linear range, correlation, LOD and LOQ for the phenolic compounds quantified. The validation parameters could be added as a Supplementary material.
Determination of Antioxidant Activity: Please add the linear range and the correlation coefficient of the Trolox curve
Please move Table 2 to material and methods section
Author Response
Reviewer 4
The article entitled: “Use of almond skins to improve nutritional and functional properties of biscuits: an example of upcycling” described the use of almond skins, an almond by-product, in the formulation of functional biscuits. The authors accessed how this addiction interfered in the proximate composition, color, friability, spread factor, phenolic profile, antioxidant activity and sensory properties. The work is well written, and it brings some interesting results and applications which are of interest to the Foods readers.
We acknowledge the Reviewer for his/her positive evaluation of our research, and we thank him/her for careful reading and for helpful suggestions.
I have some minor issues that need to be addressed:
Introduction: Line 51 : In the phrase : “The reuse of almond skins in food products would an example of upcycling [16],” something is missing, please revise it.
Response: Sorry for mistake, we corrected the sentence (see line 52).
Lane 102: The same applies for Line Energy value (kJ) was calculated by Atwater general, please revise.
Response: Sorry for mistake, we corrected the sentence (see lines 105-106).
HPLC analysis of Phenolic Compounds: The authors are invited to add the linear range, correlation, LOD and LOQ for the phenolic compounds quantified. The validation parameters could be added as a Supplementary material.
Response: Thanks for suggestion. We added a Table as Supplementary Material, containing all the suggested validation parameters.
Determination of Antioxidant Activity: Please add the linear range and the correlation coefficient of the Trolox curve
Response: We added the linear range and the correlation coefficient of the Trolox curve (see lines 154-156).
Please move Table 2 to material and methods section
Response: We moved Table 2 to Material and methods section.